# Heterogeneity of Integrin α_IIb_β_3_ Function in Pediatric Immune Thrombocytopenia Revealed by Continuous Flow Cytometry Analysis

**DOI:** 10.3390/ijms21093035

**Published:** 2020-04-25

**Authors:** Alexey A. Martyanov, Daria S. Morozova, Maria A. Sorokina, Aleksandra A. Filkova, Daria V. Fedorova, Selima S. Uzueva, Elena V. Suntsova, Galina A. Novichkova, Pavel A. Zharkov, Mikhail A. Panteleev, Anastasia N. Sveshnikova

**Affiliations:** 1National Medical Research Center of Pediatric Hematology, Oncology and Immunology named after Dmitry Rogachev, 1 Samory Mashela St, Moscow 117198, Russia; 2Center for Theoretical Problems of Physico-Chemical Pharmacology, Russian Academy of Sciences, 30 Srednyaya Kalitnikovskaya str., Moscow 109029, Russia; 3Institute for Biochemical Physics (IBCP), Russian Academy of Sciences (RAS), Russian Federation, Moscow, Kosyigina 4 119334, Russia; 4Faculty of Physics, Lomonosov Moscow State University, 1/2 Leninskie gory, Moscow 119991, Russia; 5Faculty of Basic Medicine, Lomonosov Moscow State University, 27/1 Lomonosovsky av., Moscow 119991, Russia; 6Faculty of Biological and Medical Physics, Moscow Institute of Physics and Technology, 9 Institutskii per., Dolgoprudnyi 141700, Russia; 7Department of Normal Physiology, Sechenov First Moscow State Medical University, 8/2 Trubetskaya St., Moscow 119991, Russia

**Keywords:** platelet, integrin αIIbβ3, calcium signaling, immune thrombocytopenia, blood platelet disorders, flow cytometry, computational modeling

## Abstract

Immune thrombocytopenia (ITP) is an autoimmune condition primarily induced by the loss of immune tolerance to the platelet glycoproteins. Here we develop a novel flow cytometry approach to analyze integrin α_IIb_β_3_ functioning in ITP in comparison with Glanzmann thrombasthenia (GT) (negative control) and healthy pediatric donors (positive control). Continuous flow cytometry of Fura-Red-loaded platelets from whole hirudinated blood was used for the characterization of platelet responses to conventional activators. Calcium levels and fibrinogen binding were normalized to ionomycin-induced responses. Ex vivo thrombus formation on collagen was observed in parallel-plate flow chambers. Platelets from all ITP patients had significantly higher cytosolic calcium concentration in the quiescent state compared to healthy donors (15 ± 5 nM vs. 8 ± 5 nM), but calcium increases in response to all activators were normal. Clustering analysis revealed two subpopulations of ITP patients: the subgroup with high fibrinogen binding (HFB), and the subgroup with low fibrinogen binding (LFB) (8% ± 5% for LFB vs. 16% ± 3% for healthy donors in response to ADP). GT platelets had calcium mobilization (81 ± 23 nM), fibrinogen binding (5.1% ± 0.3%) and thrombus growth comparable to the LFB subgroup. Computational modeling suggested phospholipase C-dependent platelet pre-activation for the HFB subgroup and lower levels of functional integrin molecules for the LFB group.

## 1. Introduction

Immune thrombocytopenia (ITP) is an autoimmune condition characterized by isolated low platelet count and life-threatening bleeding, primarily induced by the loss of immune tolerance to the platelet glycoproteins [1,2]. Not only platelet count but also platelet function in ITP has been reported to be affected, which can have an impact on bleeding risks, disease progression, and treatment success. Still, although abnormal platelet function in ITP was reported by many previous studies in children and adults, the significant discrepancy remains. The vast majority of reports agree that ITP platelets are pre-activated (mostly by their P-selectin staining in the resting state [3,4,5,6,7,8,9], although there are exceptions [10,11]. In contrast, the response of ITP platelets to stimulation, in particular their vitally important function of integrin α_IIb_β_3_ activation (that is necessary for aggregation and is, therefore, the immediate mechanism of primary hemostasis), is an issue of ongoing debate. Some studies report impairment [3,10], sometimes in association with bleeding risks [5,6,12], while a number of others observed normal or even increased response to stimulation [7,8,9,11,13,14,15].

Integrin α_IIb_β_3_ (glycoprotein IIb-IIIa) is the most abundant molecule on the platelet surface responsible for mediating the formation of platelet aggregates [16,17]. A disruption of blood vessel walls results in the subendothelial matrix proteins (collagen, laminin, and fibronectin) exposure [16,18] and platelet adhesion via attachment through glycoprotein Ib (GPIb) receptor and multimeric plasma protein von Willebrand Factor (vWF) [19]. The adhesion induces platelet intracellular tyrosine kinase signaling from GPIb [20], integrin α_2_β_1,_ and collagen receptor glycoprotein VI (GPVI), leading to phospholipase C (PLC) γ2 activation [21,22]. Consequently (or alternatively) soluble platelet activators like thrombin, ADP, and thromboxane A2 stimulate their G-protein coupled receptors leading to PLCβ activation [23]. This signaling involving both tyrosine kinase networks and intracellular calcium increase (mediated by PLC) gives rise to the multitude of platelet responses [24]. In particular, it causes the transition of integrin α_IIb_β_3_ into an active conformation capable of tight binding of fibrinogen and von Willebrand factor molecules [25,26]. This forms bridges between platelets responsible for platelet aggregate formation and stabilization.

The physiological importance of integrin α_IIb_β_3_ is evidenced by a genetically encoded condition called Glanzmann thrombasthenia (GT), when one of the two GP forming the integrin is defected or absent, leading to a phenotype characterized by mucocutaneous bleeding and bleeding after trauma or surgery [27,28]. The diagnostic hallmark of the disease is the lack or severe impairment of platelet aggregation induced by all agonists [29]. Genetic defects of signaling proteins regulating α_IIb_β_3_ activation give a similar phenotype [17,29]. There is also an autoimmune condition closely related to ITP and GT, which is called the acquired GT (aGT) [30] when the GT-like phenotype without thrombocytopenia is induced by autoimmune antibodies formed against α_IIb_β_3_. These antibodies in aGT could block α_IIb_β_3_ binding to fibrinogen [31], or cause mild thrombocytopenia [32], or both [33]. It could be speculated that aGT is a form of ITP with mild thrombocytopenia [30]. As a matter of fact, more than 50% of ITP patients have antibodies against α_IIb_β_3_ [34], and therefore it is possible that these antibodies could directly affect integrin function even when it is not a clear-cut aGT. In addition, platelet pre-activation in ITP (supposedly caused by some antibodies [35,36] or by complement activation [37]) may affect their integrin activation responses to stimulation via signal transduction mechanisms. Therefore, both the direct and indirect effects of ITP antibodies on platelet integrins are possible.

Here we performed analysis of platelet integrin activation and calcium signaling in blood samples of pediatric patients with ITP by means of continuous flow cytometry with additional techniques. We demonstrated that platelets from all ITP patients had significantly higher cytosolic calcium concentration in the quiescent state compared to healthy donors. Meanwhile, platelet shape change in response to ADP and relative fibrinogen binding varied significantly within the ITP group. A subgroup of ITP patients with low integrin activation altogether demonstrated phenotype similar to Glanzmann thrombasthenia. The computational modeling analysis suggested impaired integrins for this population.

## 2. Results

### 2.1. Calcium Signaling and Integrin Activation in Platelets of Immune Thrombocytopenia Patients

In order to characterize calcium signaling and functional responses of platelets in response to stimulation, continuous flow cytometry assay of diluted whole blood samples was developed for this study. The RBC (red blood cells)-settled blood sample (see Methods) was diluted to achieve the final platelet concentration of 1000–3000/μL. Platelets were identified on the forward scattering (flow cytometry channel)-side scattering (flow cytometry channel) (FSC-SSC) dot plot as a distinct cell population (Figure 1a). Calcium mobilization was calculated from changes in Fura-Red fluorescence in response to activation (Figure 1b). Cytosolic calcium concentration in quiescent platelets for healthy pediatric donors appeared to be 8.0 ± 4.7 nM, while in response to activation (ADP, TRAP (thrombin receptor activating peptide), or CRP (collagen-related peptide)) it was 50–300 nM in the maximum (Figure 1e and Appendix A). Integrin activation was assessed by immediate changes in Alexa488-labeled human fibrinogen binding from the solution (Figure 1c). Although different donors demonstrated different mean fluorescence of bound fibrinogen (450 ± 250 m.f.i.), dynamics and levels of integrin activation were assessed in relative units compared with the level of quiescent platelets taken as “0” and a stationary level achieved approximately five to ten minutes after addition of 1 μM of ionomycin as “1” (Figure 1c,f). Maximum integrin activation for healthy pediatric donors in response to stimulation (ADP, TRAP, or CRP) appeared to be 0.05–0.2 in the first five minutes of activation (Figure 1e and Appendix A). Platelet shape change in response to activation was assessed by changes in the mean area signal of SSC (SSC-A) signal of the platelet population (Figure 1d and Appendix A).

The same analysis was performed for blood samples from pediatric patients with ITP (Appendix A) and GT (Appendix A). It appeared that ranges of the possible SSC values, calcium concentrations, and fibrinogen binding was significantly wider for ITP patients in comparison to GT patients and healthy donors (ADP: Figure 1g–i, red curves; other types of activation—Appendix A). The most pronounced differences in platelet responses were noticed after stimulation with 2 μM of ADP (Figure 1). Although the ITP responses for other types of stimulation were noticeably wider, that those of healthy donors or GT (Appendix A) Platelets of ITP patients had on average larger SSC-A values (Figure 1g), reflecting their larger size in agreement with previous studies [9,38], and higher maximal calcium concentrations (Figure 1h). SSC-A values and calcium concentrations of GT and healthy donors corresponded to each other (Figure 1g,h, blue and green curves). Expectedly, GT patients had impaired fibrinogen binding (Figure 1i). However, although platelets from ITP patients on average had fibrinogen binding comparable to healthy donors, at least some samples from ITP patients had fibrinogen binding comparable to samples from GT patients (Figure 1i and Figure 2a).

### 2.2. Clustering Analysis of Flow Cytometry Data Reveals Two Subgroups of Patients with ITP

The wide range of platelet responses to activation in ITP patients (Figure 2a) induced us to search for subgroups by means of clustering analysis. The patients were clustered as dots with relative values of the following parameters as coordinates: the calcium concentration in quiescent state ([Ca^2+^]_Basal_), maximal calcium concentration ([Ca^2+^]_Max_), relative fibrinogen binding and relative SSC-A change upon activation with 2 µM of ADP (Figure 2b,c). The same analysis was performed for other types of stimulation (Appendix A), but the statistically significant division into subgroups was achieved only for ADP-induced activation. It appeared that ITP patients are best divided into two groups, mostly distinguished by their dramatically different ability to bind fibrinogen in response to stimulation: the high fibrinogen binding (ITP: high fibrinogen binding (HFB); 67% male, 36% female patients, blue circles, *n* = 12) group and the low fibrinogen binding (ITP: low fibrinogen binding (LFB); 45% male, 55% female patients, green circles, *n* = 21) group.

We performed a comparison of platelet responses for ITP patients with those of pediatric GT patients (n = 8) and healthy pediatric donors (*n* = 14). Both subgroups of ITP appeared to have a not statistically significant increase in platelet size (SSC-A mean values: 70 ± 10 × 10^2^ (Healthy), 73 ± 9 × 10^2^ (GT), 82 ± 15 × 10^2^ (ITP:LFB), and 83 ± 21 × 10^2^ (ITP:HFB)) and cytosolic calcium concentration in a quiescent state (Figure 2c,d) in comparison to GT patients and healthy donors. Upon activation with ADP platelets from ITP:LFB group and GT demonstrated 20%–25% shape change, while platelets from ITP:HFB group and healthy donors had significantly higher shape change of 30%–32% (Figure 2c,f). On the other hand, maximal calcium concentration was higher for ITP:HFB group (155 ± 27 nM) than in ITP:LFB group (101 ± 31 nM), GT (81 ± 23 nM) or healthy donors (91 ± 22 nM) (Figure 2b,e). Finally, although being significantly weakened, fibrinogen binding to platelets from ITP:LFB patients (8.2% ± 4.8%) was higher than from GT patients (3.4% ± 0.9%), while fibrinogen binding to platelets from ITP:HFB patients (21% ± 5%) was comparable to values demonstrated by healthy donors (16% ± 4%) (Figure 2b,g). Based on these observations, it can be claimed that platelet activation in the ITP:LFB group could be compared to that of GT. The same characteristics for other types of stimulation for the ITP:HFB and the ITP:LFB groups are given in Appendix A. Although the difference between the subgroups for other types of stimulation is less distinct, the same trend in integrin activation, calcium mobilization, and shape change is preserved.

### 2.3. Computational Models of Integrin Activation for ITP Subgroups

In order to understand whether the increased calcium concentration in a quiescent state is linked to integrin activation, we developed a computational systems biology model of platelet calcium signaling and integrin activation (Figure 3a) based on our previously published models on platelet intracellular signaling [39,40,41,42]. The model had all compartments and major calcium signaling mechanisms. Thus, integrin activation was added as a function of Rap1 binding to GTP, which was assumed to be influenced by activation of calcium diacylglycerol guanidine exchange factor I (CalDAGGEFI) and phosphoinositide-3-kinase (PI3K)-dependent inactivation of Rap1 GTPase activating protein (Rap1GAP) RASA3. PI3Kγ was assumed to be activated by Gβγ released from the ternary complex primarily by active P2Y_12_ receptor (see Appendix A for model equations), while PI3Kβ was activated by the tyrosine-kinase signaling cascade started by GPVI receptor.

We propose two possible reasons for the increased cytosolic calcium concentration in the quiescent state of platelets of ITP patients.

The first is an increased membrane conductivity for calcium ions due to the activation of some calcium channels (for ex. P2X_1_) or due to smaller average age of platelets. We modified the initial model by increasing the intracellular calcium stores membrane conductivity (Figure 3b,c, LFB model), so that basal calcium increased from 10 to 15 nM. Surprisingly, an increase in plasma membrane calcium conductivity lead to a HFB-like phenotype (Appendix A). Upon activation, this model demonstrated the same calcium mobilization and fibrinogen binding in response to ADP as the initial model (Figure 3b,c, N model). The calcium response resembled that of the ITP:LFB group, while the model could be tuned to describe the diminished integrin activation only if the amount of one of the signaling proteins is lowered (Figure 3c, LFB* model—the amount of Rap1 is 75% of normal). Therefore, we speculate that the integrin molecules themselves must be affected by the autoantibodies in this group.

The second proposed mechanism for the elevated cytosolic calcium concentration in the quiescent state of platelets is the pre-activation of platelets due to the binding of antibodies to some G-protein coupled or associated with tyrosine kinases receptors, leading to PLC activation and an increased level of IP_3_ in the cytosol. We modified the model by supplementing it with PLC activity in a quiescent state (Figure 3d, Appendix A, HFB model) so that basal calcium increased from 10 to 15 nM. Upon activation this model demonstrated increased calcium mobilization and fibrinogen binding in response to ADP (Figure 3d) and CRP (Appendix A) compared to the initial model (Figure 3d, Appendix A, N model). These responses closely resembled those of the ITP:HFB group. Thus we speculate that platelets from these patients could be pre-activated by some antibodies [36].

### 2.4. Ex Vivo Thrombus Growth and Leukocyte Activity for ITP Subgroups

Although platelets from the ITP:HFB subgroup appear to be pre-activated and to have an enhanced response to stimulation than healthy donors, patients of this subgroup still had decreased platelet count (78 ± 19 × 10^9^ plt/L on average) and bleedings (Buchanan bleeding score 0.81 ± 0.26 on average). Thus, we have checked the capability of the platelets of this group to form thrombi ex vivo under a low shear rate (Figure 4a). Platelets of ITP:LFB, GT, and healthy groups (Figure 4b–d) were analyzed in the same conditions. Blood was collected from ITP donors without severe thrombocytopenia (plt > 30 × 10^9^/L). Thrombus area after 20 min was lower both for ITP (3% ± 2% for HFB, 10% ± 4% for LFB) and for GT patients (10% ± 3%) compared to healthy donors (13% ± 6%). Strikingly, the thrombus area was decreased more significantly in ITP:HFB. Furthermore, the decrease in thrombus area in ITP:LFB and GT patients was not as significant as it could have been expected based on their bleeding scores (Figure 4e). To test the potential impact of low platelet count in ITP patients, we have performed experiments with whole blood from healthy donors with five-times reduced amount of platelets (see Methods). It appeared that the thrombus area, as well as thrombus growth parameters (Table 1) in this conditions, was comparable to the ITP:LFB group (Figure 4f).

Because the area of thrombi appeared to depend on the platelet count critically, we have analyzed the heights of thrombi and the rate of thrombus growth (Table 1). We assumed that 3,3’-Dihexyloxacarbocyanine Iodide (DiOC6) intensity corresponds to the thrombus height. It appeared that although thrombus areas of ITP:LFB and GT patients are close to being normal, the rate of growth of the thrombus in height above the collagen surface was impaired in these patients (1.51 ± 0.50 and 1.45 ± 0.40, correspondingly) in comparison to healthy donors (2.7 ± 0.3), but not to healthy donors with low platelet count (1.47 ± 0.29). Finally, thrombi in samples from ITP:HFB subgroup were growing most effectively (3.5 ± 0.9). Furthermore, when we calculated the fraction of individual thrombi those proceed to grow in height and area after 20 min of the experiment, this parameter was also lower both in ITP:LFB (44% ± 15%) and GT patients (60% ± 27%) in comparison to healthy donors (74 ± 26%) and ITP:HFB subgroup (72% ± 36%). Typical non-growing and growing thrombi are given in Appendix A.

The ability of growing thrombi to attract leukocytes by the released platelet granule contents as a chemoattractant could also be used as a parameter of platelet activation [43,44,45]. Under low shear rate conditions, PMNs (polymorphonuclear neutrophils) crawling among the growing thrombi can be observed [46]. The PMN crawling velocity of the ITP:LFB patients was decreased at the beginning of the experiment (48 ± 22 nm/s vs. 80 ± 20 nm/s for healthy controls), but increased to normal values with time (73 ± 23 nm/s vs. 76 ± 22 nm/s for healthy controls) (Appendix A). Other differences between PMN velocities, although being significant, were less pronounced (Appendix A). This demonstrates that platelets’ activation during thrombus formation is normal for both ITP subgroups.

## 3. Discussion

In this study, we develop a continuous flow cytometry approach to quantify integrin α_IIb_β_3_ activation dynamics by fibrinogen binding and cytosolic calcium mobilization upon stimulation of human platelets. With it, we show that platelets of pediatric ITP patients reveal a clear heterogeneity in their signaling and integrin activation dynamics so that they form two subgroups that could be clinically significant.

Continuous flow cytometry approach described here allows simultaneous assessment of the platelet intracellular signaling (cytosolic calcium concentration) alongside primary functional responses (size, shape change, phosphatidylserine exposure, and fibrinogen binding) both at the quiescent state and upon activation (Figure 1) upon strong dilution of the sample to make it independent of the platelet count and secondary activation. Although several approaches for platelet calcium signaling measurement on flow cytometer were proposed previously [39,47,48,49], the technique utilized here allows precise quantitative measurement of calcium concentration independent of platelet size and activation state because of the utilized mild ionomycin and EGTA concentrations and a direct comparison of calcium concentration in a quiescent state with an EGTA-provided mark of 50 nM. Our study is the first to our knowledge to evaluate integrin activation in ITP by using the native integrin α_IIb_β_3_ ligand, fibrinogen. Although for human platelets, PAC-1 antibody binding is the first choice for flow cytometry [5,8,50], it has several well-known drawbacks such as slow binding [51] and sensitivity to the high activation state [51,52]. The patients in our study group were children, with relatively same proportions of persistent and chronic ITP, bleeding, and non-bleeding ones (Appendix A).

In this study, we agree with the view held by the majority of previous studies [4,5,6,7,8,9,13] that platelets in ITP are pre-activated. In contrast to others, we, for the first time, observe it not by their exterior parameters but by increased resting intracellular calcium. Using the Hierarchical Density-based spatial clustering of applications with noise (H-DBSCAN) clustering algorithm, two subgroups of ITP patients were distinguished not only by the level of fibrinogen binding but by maximal calcium concentrations and shape change (Figure 2b,c). These differences were significant in response to ADP (Figure 1), while differences in response to CRP were less pronounced, although still detectable (Appendix A). This discrepancy could be caused by the variability of platelet reactivity to CRP [53] based on the variable number of GPVI receptors [54], GPVI receptor polymorphisms [55,56] or other differences [53,57,58]. The systems biology analysis of intracellular signaling in HFB and LFB subgroups proposed a significant difference in the origin of their pre-activation. For the HFB group the true pre-activation possibly due to GP1b activation by the antibodies [35] leading to Syk-mediated PLCγ2 activation [24] and constant elevation of IP_3_ platelet level (Figure 3c). Our preliminary data confirm that ITP plasma could cause pre-activation observed here (Appendix A). It appeared that while pre-incubation with healthy donor plasma did not alter platelet reactivity to 2 µM ADP, a pre-incubation with the plasma of one of the three ITP donors resulted in the significantly increased calcium and fibrinogen responses to ADP (Appendix A). This kind of pre-activation could lead to P-selectin exposure observed in previous studies. For the LFB group, a “false” pre-activation was suggested based on the increased membrane calcium conductivity without altering of the platelet responses to activation (Figure 3d). This type could be caused by the altered platelet lifespan [59] or platelet interactions with the immune system [36,37].

Integrin α_IIb_β_3_ activation in response to stimulation with ADP, TRAP, or CRP clearly identifies two distinct groups of comparable size with dramatically different fibrinogen binding. One is comparable to healthy donors (who can be considered positive control), while another was similar to Glanzmann’s thrombasthenia patients (who could be likewise considered negative controls) (Figure 2). To our knowledge, this is the first observation of clear two-peaked distribution in ITP platelet functionality (Figure 2a). In contrast to previous reports using PAC-1, fibrinogen binding in our study did not correlate with clinical bleeding (Appendix A), nor did they anti-correlated with platelet pre-activation [6], though it was associated with the difference in thrombus growth dynamics (Table 1). Together with no apparent defect in signal transduction, with the independence of the groups of the platelet stimulation method (Figure 2 and Appendix A), and with their independence of the disease stage, therapy or other parameters (Appendix A and Appendix A), this suggests that fibrinogen binding defects observed here could be due to impaired functioning of integrins themselves, e.g., with direct action of antibodies against α_IIb_β_3_, which are indeed discovered in approximately 50% of the ITP patients [60,61].

The two subgroups of ITP patients identified here were similar by Buchanan bleeding score (0.81 ± 0.26 for HFB and 0.90 ± 0.15 for LFB) and average platelet count (90 ± 21 plt/nL for LFB and 78 ± 21 plt/nL for HFB). Lack of association of fibrinogen binding with clinical bleeding could be misleading because the majority of bleeding observed in thrombocytopenia is spontaneous bleeding caused by issues with vascular integrity [62,63]. They are not necessarily associated with traditional platelet functions like aggregate formation because sealing of the breaches in the vasculature can seemingly be done by single platelets [64,65]. We endeavored to compare ex vivo thrombus formation on collagen for several patients from these groups (Figure 4). Surprisingly, the average area of thrombi coverage for HFB group was lower than for LFB group, demonstrating that platelet count is more crucial for the size of thrombi than the functioning of platelets. By means of the thrombus growth assays, we have identified additional similarities between ITP:LFB and GT groups: average thrombus areas of both groups were not significantly decreased in comparison to healthy donors. However, in both LFB and GT groups, thrombus growth above the first layer of platelets was significantly impaired (Figure 4 and Table 1), while the thrombus growth rate of the HFB group was even higher than for the healthy donors (Figure 4e and Table 1). Based on these data, it could be claimed that the activity of platelet integrins determines the rate of thrombus growth, rather than the final thrombus area, thus explaining relatively higher bleeding scores of LFB group of patients. This relationship between platelet functioning and integrin activation is in line with theoretical models of thrombus formation [66,67] as well as in vivo studies on murine models [68,69].

The present study suggests potential activatory and inhibitory roles for autoantibodies present in ITP plasma. The preliminary experiments performed here (Appendix A) partly supports this theory. However, further experiments on integrin functioning in ITP, for example, their ability to adhere to fibrinogen, as well as direct measurement of the autoantibody content by such methods as MAIPA (Monoclonal Antibody Immobilization of Platelet Antigens) [60,61] are necessary to examine this point and will be the subject of future research.

Taken together, our data cautiously suggest a novel view of the ITP platelet functioning. According to this view, the ITP platelets have normal signal transduction despite being pre-activated, yet approximately 50% of patients have severely (to the degree comparable with Glanzmann’s thrombasthenia) impaired fibrinogen binding in response to stimulation associated with defects in thrombus formation under flow. We can very carefully speculate that these defects could be due to the direct action of antibodies against integrin α_IIb_β_3_.

## 4. Materials and Methods

### 4.1. Patients

Thirty-six patients aged 2 to 17 years diagnosed with immune thrombocytopenia (Appendix A), 6 patients aged 2 to 10 years with 2 adult parents diagnosed with Glanzmann thrombasthenia and 8 healthy pediatric donors (age 2−14 years, 50% male and 50% female) with three adult donors were included in the study. Investigations were performed in accordance with the Declaration of Helsinki under a protocol approved by the “Independent CTP PCP RAS Ethical Committee” (CTP PCP RAS, Srednyaya Kalitnikovskaya str., Moscow 109029, Russia; protocol #1 from 12.01.2018), and written informed consent was obtained from all donors and patients. Primary ITP was diagnosed on the basis of isolated thrombocytopenia (platelet count below 100 × 10^9^/L) with secondary thrombocytopenia excluded, according to the American Society of Hematology (ASH) Guidelines [70]. Glanzmann thrombasthenia was diagnosed on the basis of commonly accepted criteria.

### 4.2. Materials

The sources of the materials were as follows: calcium-sensitive cell-permeable fluorescent dye Fura-Red-AM, (Molecular Probes, Eugene, OR); ADP, PGI2, EGTA, HEPES, bovine serum albumin, apyrase grade VII, ionomycin, SFLLRN (Sigma-Aldrich, St Louis, MO, USA); CD66b-PE (Sony Biotechnology, San Jose, USA); Cysteine-containing version of cross-linked collagen-related peptide (CRP) was kindly provided by Prof. R.W. Farndale (University of Cambridge, Cambridge, UK). Tyrode’s buffer (150 mM NaCl, 2.7 mM KCl, 1 mM MgCl2, 2 mM CaCl2, 0.4 mM NaH2PO4, 0.4 mM Na2CO3, 5 mM HEPES, 5 mM glucose, 0.5% BSA, pH 7.35) was fresh made from reagents (Sigma-Aldrich, St Louis, MO, USA).

### 4.3. Continuous Flow Cytometry Assessment of the Platelet Intracellular Signaling and Functional Parameters

For the flow cytometry, hirudinated whole blood was incubated with 2 µM of Fura-Red in the presence of 1 U/mL of apyrase for 35 min at 37 °C. The leukocyte rich plasma (LRP) was collected above the settled red blood cells and diluted in the Tyrode’s-calcium buffer to the final concentration of 1000 plt/mL and left resting for 30 min. 100 µL of LRP was the lowest dilution used in our conditions. In order to check the potential effects of different concentrations of blood plasma present in the system, we have performed preliminary experiments with differently diluted LRP of healthy donors (Appendix A). It appeared that at 50 times (30 µL LRP per 1530 µL Tyrode’s) and at 25 times (60 µL of LRP per 1530 µL Tyrode’s) no effect of the increased plasma and platelet concentration was present (Appendix A d–f). However, at 15 times (100 µL LRP per 1530 µL Tyrode’s) dilution, platelet reactivity appeared to be increased, yet this effect was not statistically significant (Appendix A). Based on our additional experiments, this is based not on the increased plasma concentration, but on the increased platelet concentration, at which secondary platelet activation upon secretion becomes more pronounced [42,71,72]. To test the potential impact of the antiplatelet antibodies in ITP patient plasma, we have pre-incubated platelets with Tyrode’s buffer, healthy donor platelet free plasma or ITP donor platelet free plasma under stirring conditions for 30 min. Platelet free plasma for the experiments with plasma pre-incubation was collected upon sequential centrifugation at 250 g for 15 min and 2000 g for 15 min in from the citrated whole blood and stored at −80 upon being frozen in liquid nitrogen. Then,100 µg/mL of Alexa-488 labeled human fibrinogen was added 2 min before sample loading to the BD FACS Canto II Flow Cytometer (BD Biosciences, San Jose, CA, USA). Samples were analyzed in a continuous mode. The typical whole blood FSC-SSC region, corresponding to platelets, is given in Figure 1a.

The primary fluorescence signal was averaged over equal periods (1s, blue curves in Figure 1b–d). The ratio of calcium bound Fura-Red (excited by 405 nm laser, Figure 1b) to calcium-free Fura red (excited by 488 nm laser, Figure 1b) was recalculated into platelet cytosolic calcium concentration using Grynkiewicz formula [73]:(1)$$\left[{\mathrm{Ca}}_{\mathrm{Free}}^{2+}\right]{=K}_{D}^{\mathrm{furared}}\;\times \;\frac{{\mathrm{FI}}_{\max}}{{\mathrm{FI}}_{\min}}\;\times \;\frac{{R-R}_{\min}}{R_{\max}-R}$$
where $K_{D}^{furared}$ denotes Fura-Red affinity for free Ca^2+^ ions; *FI_max_* denotes the maximal fluorescence intensity for calcium-free Fura-Red, *FI_min_* denotes the minimum fluorescence intensity for calcium-free Fura-Red, *R* denotes the ratio of calcium-bound to calcium-free Fura-Red intensity, *R_max_* denotes the maximum possible value of *R*, and *R_min_* denotes the minimum possible value of *R*. *FI_min,_* and *R_max_* were obtained upon addition of 1 µM of ionomycin. Due to the inability of 10 mM of EGTA to chelate all of the free calcium ions present in ionomycin permeabilized platelets [74], calcium concentration obtained from equation (1) was normalized on the value of calcium concentration in the buffer upon EGTA addition to free calcium concentration in the presence of 2 mM of calcium and 10 mM of EGTA calculated from [74]. Fibrinogen binding was considered to be 0 % at the initial moment and 100 % upon 10-min incubation with 1 µM of ionomycin (Figure 2). The relative change in the SSC-A signal was considered to be characteristic of the platelet shape change [75].

### 4.4. Ex vivo Thrombus Growth Analysis

Ex vivo thrombus growth analysis was conducted as described previously [46,66]. Briefly, hirudinated whole blood was preincubated with DiOC-6 (500 nM) and perfused through the parallel plate flow chambers with fibrillary collagen (100 µg/mL) covered glass at 100 s^-1^ shear rate. Blood with “low platelet count” was obtained after the RBC settling for 1h, LRP collection, and supplementation of 80% of LRP by Tyrode’s calcium buffer with hirudin. Diluted LRP was then reintroduced to the whole blood and incubated for 30 min. Comparison of Tyrode’s supplemented blood and platelet free plasma supplemented blood did not provide any significant differences (data not shown). Thrombus growth was observed using a Nikon Eclipse Ti-E inverted microscope (Tokyo, Japan). Ten fields of view (FOV) were monitored during the first 10 min of the experiment and then, 10 different FOVs were monitored during the following 10 min of the experiment. Polymorphonuclear leukocytes (PMNs) were determined by diffuse DiOC-6 labeling, distinctive from typical thrombi labeling (see Appendix A).

### 4.5. Computational Modeling

A systems biology model of platelet calcium signaling was based on the ones developed previously [39,41,42]. In contrast to its predecessors, which were pure calcium signaling/homeostasis models, it contained an integrin activation module (Appendix A). Briefly, the model described activation by the main G-protein coupled receptors (ADP receptors P2Y_1_, P2Y_12_, thrombin receptors PAR1 and PAR4, but not receptor to thromboxane A2) and GPVI receptor. They all trigger phospholipase C activation. In response to IP_3_ production by PLC, free calcium ions are released from the intracellular stores, the dense tubular system (DTS) of platelets. Additionally, a phosphoinositide-3-kinase (either γ or β isoform) becomes activated, leading to an increase in PIP_3_ concentration in the plasma membrane. The increase in cytosolic calcium concentration results in small GTPase Rap1 binding to GTP by the calcium diacylglycerol guanidine exchange factor I (CalDAGGEFI) [26]. Simultaneously, the increase in PIP_3_ concentration leads to inhibition of a Rap1 GTPase activating protein (Rap1GAP), further increasing Rap1GTP concentration [76,77]. Rap1 leads to α_IIb_β_3_ transition to the active state at which α_IIb_β_3_ affinity to blood plasma protein fibrinogen is significantly increased (“inside-out” integrin activation). α_IIb_β_3_-fibrinogen- α_IIb_β_3_ bridges formation ensures thrombus growth [78].

The model parameters were adjusted to describe calcium signaling and integrin activation upon platelet stimulation with ADP or CRP at several concentrations (see Appendix A). Validation of the model was performed by the following experiment. The model predicted synergy in integrin activation between ADP and TRAP (Appendix A, right). We conducted the continuous flow cytometry experiment for platelets from healthy donors with sequential activation with ADP and TRAP in the same setting as in Figure 1. While calcium mobilization in response to TRAP after ADP did not change, integrin activation increased twice in excellent agreement with model prediction (Appendix A, left). The equations and parameters of the integrin module are given in Appendix A.

### 4.6. Data Analysis

All data analysis was performed using Python 3.6. Flow cytometry data clustering was analyzed using the H-DBSCAN clustering algorithm [79]. Statistical significance of division into clusters was assessed DBCV (Density-Based Clustering Validation) criteria [80]. PMN crawling velocity and thrombus areas were assessed using trackpy python library (DOI: 10.5281/zenodo.3492186). Thrombus intensity was assessed manually by means of ImageJ. Thrombus was considered growing if in 2 min its fluorescence has increased at least 1.1 fold.

## Figures and Tables

**Figure 1 ijms-21-03035-f001:**
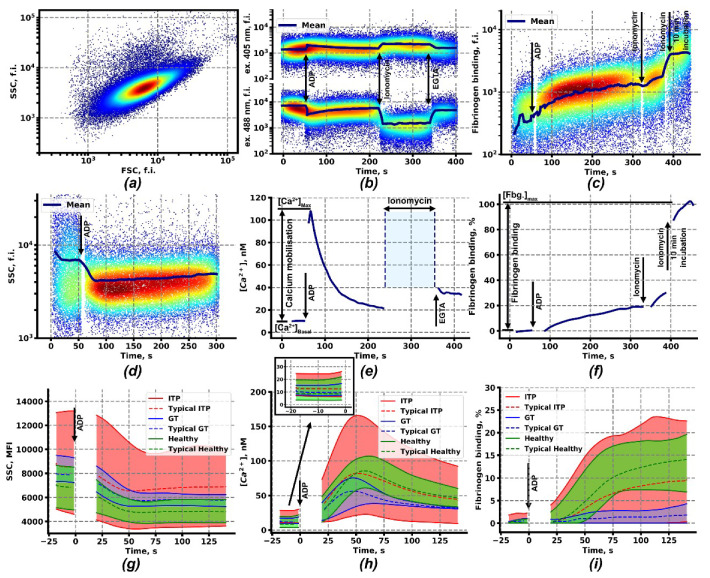
Continuous flow cytometry-based assessment of intracellular calcium concentration, fibrinogen binding and shape change in platelets upon activation. (**a**) Typical forward scattering (flow cytometry channel)-side scattering (flow cytometry channel) (FSC-SSC) dot-plot of the diluted whole blood sample; (**b**) Fluorescence of the calcium-bound (excitation 405 nm) and calcium-free (excitation 488 nm) Fura-Red upon sequential addition of ADP (2 µM), Ionomycine (1 µM) and ethylene glycol-bis(β-aminoethyl ether)-N,N,N′,N′-tetraacetic acid (EGTA, 10 mM); (**c**) Fluorescence of associated with platelets Alexa-488 labeled fibrinogen upon sequential addition of ADP (2 µM) and Ionomycine (1 µM) with further incubation for 10 min; (**d**) SSC of platelet suspension upon activation by ADP (2 µM); (**e**) Cytosolic calcium concentration (e) in platelets, calculated from Fura-Red fluorescence (b); (**f**) Relative fibrinogen binding (f) by platelets, calculated from the fluorescence of Alexa-488 labeled fibrinogen (c). (**g**−**i**) Envelopes of the mean SSC (**g**), cytosolic calcium (**h**) (with inset showing the quiescent state calcium levels), and fibrinogen binding (**i**) curves of patients with Immune thrombocytopenia (ITP) (*n* = 36, red curves), Glanzmann thrombasthenia (*n* = 8, blue curves), and healthy donors (*n* = 14, green curves) upon activation by ADP (2 µM).

**Figure 2 ijms-21-03035-f002:**
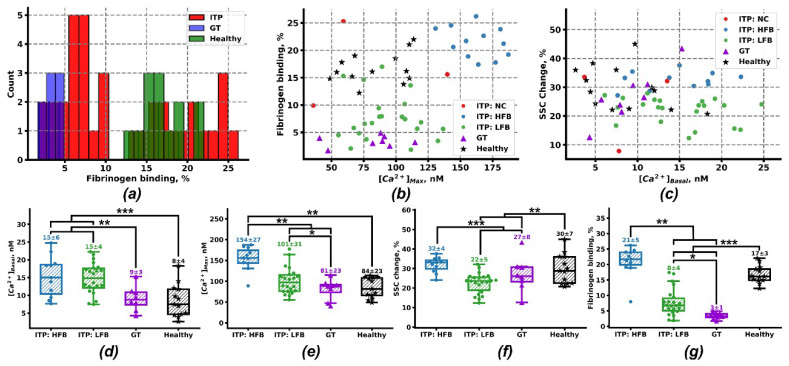
Identification and characterization of subgroups of ITP patients. (**a**) Histogram of fibrinogen binding to platelets of patients with ITP (ITP, red), Glanzmann thrombasthenia (GT, blue) and healthy donors (green). (**b**,**c**) Cluster analysis of ITP patients by H-DBSCAN (Hierarchical Density-based spatial clustering of applications with noise) clustering algorithm with calcium concentration in quiescent state ([Ca^2+^]_Basal_), maximal calcium concentration ([Ca^2+^]_Max_), relative fibrinogen binding and relative SSC-A change upon activation with 2 µM of ADP as coordinates revealed two subgroups, high fibrinogen binding (ITP: high fibrinogen binding (HFB), *n* = 12, blue circles) and low fibrinogen binding (ITP: low fibrinogen binding (LFB), *n* = 21, green circles). Three patients did not adhere to any of the groups (ITP: NC, red circles). The same characteristics for GT (purple triangles) and healthy donors (black stars) are given for comparison purposes. (**d**−**g**) Mean values ± SEM for the analyzed groups upon platelet stimulation with ADP for maximal calcium concentration (**d**), basal calcium concentration (**e**), shape change (**f**), and fibrinogen binding (**g**). Statistics was calculated by Mann-Whitney test, * *p* < 0.05, ** *p* < 0.01, *** *p* < 0.001.

**Figure 3 ijms-21-03035-f003:**
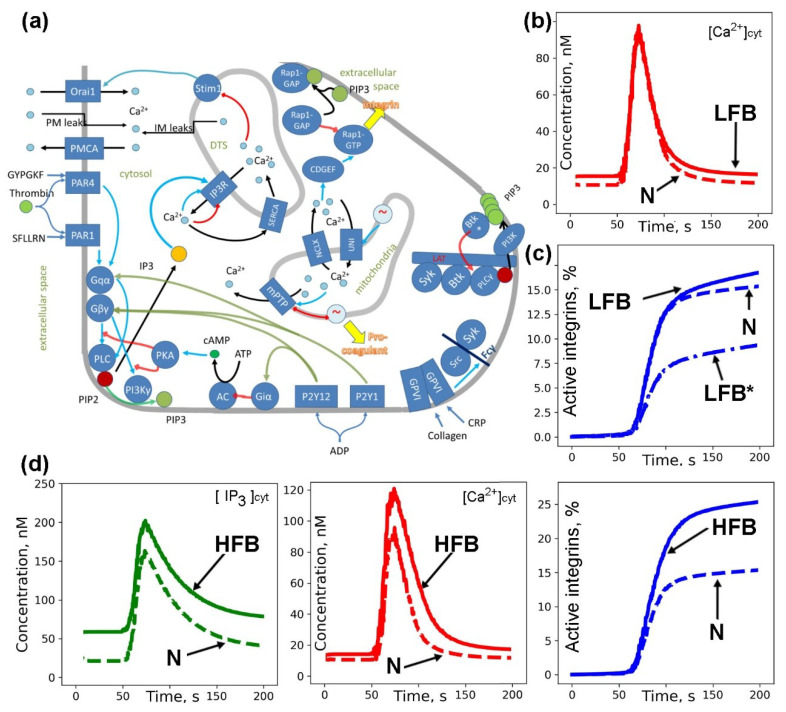
Increased cytosolic calcium as a result of platelet pre-activation or as a result of increased membrane calcium conductivity: computational models. (**a**) Scheme of signaling events. Platelet activation starts with a soluble ligand (thrombin, ADP or CRP) binding to a receptor on the platelet plasma membrane (blue rectangles). The activated G-protein coupled receptor induces heterotrimeric G-protein dissociation into α and βγ subunits. Gα_q_ and Gβγ lead to phospholipase C (PLC) activation, while Gβγ also leads to phosphoinositol-3-kinase (PI3K) activation. Gα_i_ leads to adenylate cyclase (AC) and protein kinase A (PKA) inhibition. CRP binds to GPVI receptor and induces its receptor clustering and their phosphorylation by half-active tyrosine kinases (Src), leading to binding and full activation of Syk and Src. Active kinases lead to PI3K and PLCγ activation. PLC activation leads to an increase in cytosolic free Ca^2+^ concentration. Both Ca^2+^ and PI3K lead to integrin activation through induction of small GTPase Rap1 association with GTP. Abbreviations. PAR—protease-activated receptor, P2Y—purinergic receptor, SERCA—sarcoplasmic/endoplasmic reticulum calcium ATPase, PMCA—plasma membrane calcium ATPase, CDGEF CalDAGGEFI, DTS—dense tubular system, Mit a mitochondrion, IP3R—receptor for inositol-1,4,5-trisphosphate (IP3), PIP2—phosphoinositol-4,5-bisphosphate, PIP3—phosphoinositol-3,4,5-trisphosphate, UNI—mitochondrial uniporter, NCLX—mitochondrial sodium/calcium exchanger, mPTP—mitochondrial permeability transition pore. (**b**,**c**) A computational model for platelet with increased membrane conductivity for calcium (LFB). Initial IP3 concentration (not shown) was assumed to be the same as in the initial model, while higher basal calcium concentration ((**b**), solid red) was sustained by increased calcium influx through the membrane without significant integrin activation ((**c**), solid blue). Stimulation with ADP (2 μM at 50 s) lead to the same maximal calcium concentration and integrin activation as in the initial model (N, dashed curves). In the case when one of the signaling protein is damaged, the integrin activation becomes lower (LFB* model, dash-dot curve) (**d**) Computational model for pre-activated platelet (HFB). Initial IP3 concentration (solid green) was assumed to be higher due to pre-activation, which lead to higher basal calcium concentration (solid red) without significant integrin activation (solid blue). Stimulation with ADP (2 μM at 50 s) lead to higher maximal calcium concentration and integrin activation compared to the initial model (N, dashed curves).

**Figure 4 ijms-21-03035-f004:**
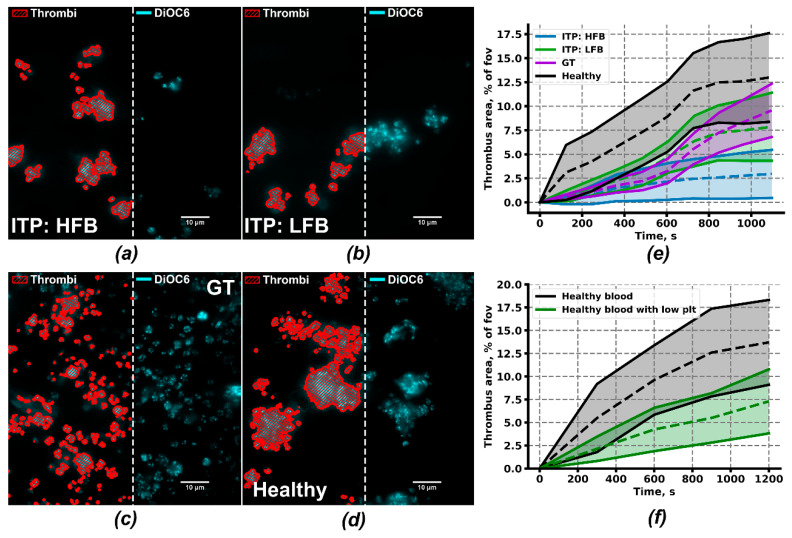
Ex vivo thrombus growth under low shear rate conditions. (**a**−**d**) Typical thrombi on fibrillary collagen of ITP:HFB (*n* = 3, **a**), ITP:LFB (*n* = 3, **b**), GT (*n* = 4, **c**) and healthy donors (*n* = 7, **d**) after 15 min of hirudinated whole blood perfusion. Method of thrombi area calculation is demonstrated by red dashed areas on the left halves of the panels. (**e**) Mean thrombus area (dashed lines) ± SD (solid lines) of ITP:HFB (blue), ITP:LFB (green), GT (purple) and healthy donors (black). (**f**) Mean thrombus area (dashed lines) ± SD (solid lines) of healthy donors (black) and healthy donors with 5-times reduced platelet count (green).

**Table 1 ijms-21-03035-t001:** Parameters of ex vivo thrombus growth after 20 min of the experiment.

Parameter	ITP:HFB	ITP:LFB	GT	Low Plt^1^	Healthy
Fraction of thrombus area, %	3 ± 2^***2,3^	10 ± 4^*^	10 ± 3^*^	7 ± 3^**^	15 ± 4
Fraction of continuously growing individual thrombi	72 ± 26^ns^	44 ± 15 ^ns^	59 ± 27 ^ns^	60 ± 14 ^ns^	74 ± 36
Maximal thrombus growth, fold change	3.5 ± 0.9 ^ns^	1.5 ± 0.5^**^	1.5 ± 0.4^**^	1.47 ± 0.29^*^	2.7 ± 0.3
Growth velocity, %/min	25 ± 12 ^ns^	9 ± 5 ^ns^	7 ± 5 ^ns^	7 ± 4^*^	17 ± 8

^1^ Low Plt – samples from healthy donors with 5-times reduced platelet count. ^2^ SD values for *n* = 30 thrombi. ^3^ Statistical significances in comparison to Healthy donors was calculated by Mann–Whitney test, * *p* < 0.05, ** *p* < 0.01, *** *p* < 0.001, ns – non significant.

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
