# Peer review of "Heterogeneity of Integrin α_IIb_β_3_ Function in Pediatric Immune Thrombocytopenia Revealed by Continuous Flow Cytometry Analysis"

_ijms, 2020, doi:10.3390/ijms21093035_

Round 1

Reviewer 1 Report

The manuscript by Martyanov et al provides with data of a2Bb3 integrin activation (through fibrinogen binding), calcium flux and morphological platelet variables in resting state and upon stimulation with various agents, in a cohort of pediatric ITP patients, in comparison with healthy controls and Glanzmann Thrombasthenia patients (age matched).

While the study is very interesting, there are some issues that would require revision, or additional experiments.

The first general concern, is the fact that the blood is diluted to perform experiments, even when the same concentration of platelets is set as the dilution justification, the dilution factor is not comparable between patients, and thus, the final sample might be altered by that (and responses), considering a different dilution of RBCs and plasma in the mixture...

Experiments should be done with whole blood, or reconstituted blood (to have the same amount of platelets per volume unit), or with washed platelets, with monitored plasma presence, etc.

One of the main conclusions of this work, is that ITP platelets are preactivated, compared to healthy donor platelets. I have my doubts about this conclusion, since in Figure 1, it is shown that the SSC before stimulation with ADP (or agonist) is higher already in ITP platelets (wider range as well)... suggesting that actually, the integrity of ITP platelets is better preserved than those from healthy donors, which would also fit with a limited reaction capacity. The interpretation of basal Ca results are also a bit confusing, as graphs in Figure 1h do not entirely correspond with graphs in Figure 2d.

Please discuss about this and reconsider the conclusion.

Figure 2 shows the stratification of ITP platelets on two subgroups based on their ability to bind FBG, which coincides with a reduced change on SSC.

  • What is the basal SSC on the two subgroups of ITP?, is it already different as to explain a potential "pre-activation" and anergy in one of the subgroups? Or is it not altered? The second would also support the explanations given by the authors.
  • Figure 2c, the legend covers some of the points.

The computational model goes beyond my evaluation capacity, but I would be cautious with some statements, such as "increased cytosolic calcium concentration in quiescent state of platelets, and its explanation - could be preactivated with antibodies, or permeability": it is not clear to me whether Ca2+ is actually increased, and the assumption of antibodies causing pre-activation is not proven at all, neither the permeability.

  • An experiment to try to prove this, would be to incubate healthy donor platelets with plasma/serum/Isolated Ig Fraction from the pediatric ITP patients, to see if that pre-activation could be induced.
  • Permeability could be assessed by permeable dies.

While the stratification of ITP platelets, based on fibrinogen binding, was found (as said in the text) only significant with ADP, the ex vivo thrombus formation experiment is performed on flow chambers coated with collagen.

Still, in Supplementary, it looks as if stimulation with CRP does separate significantly those ITP subgroups based on FBG binding. Still, CRP seems to work through GPVI activation, and collagen might be recognized more efficiently by B1 integrin.

Please explain about this, and add Syk as another factor on the computational model.

  • The flow assay could be done with fibrinogen coated chambers.

Another caveat of this assay is the fact that whole blood was assessed, with the difference on platelet counts. As mentioned before, it should be done with the same number of platelets per volume unit, which can be done by reconstituting blood (not diluting), or by performing the assay with washed platelets, with a specific plasma proportion.

The fact that we always have patient plasma on the samples that are tested, is a limitation on the interpretation of data (is something in the plasma interfering with results?, which results are platelet-dependent or endogenous to platelets?). This should be addressed by the authors in the disunion, and some example experiments should be performed to show the results obtained are platelet-dependent or not.

I find absolutely necessary to provide data on autoantibodies in these patients, MAIPA testing or similar, to identify presence or not of B3 antibodies (vs antibodies against other glycoproteins). A correlation of results, with these valuable data, could provide with a better explanation of the findings.

Regarding the Supplementary data, the tables do not contain the platelet measuring unit. It would be great to have IPF on those samples, and the same table for the healthy donors. What is TPO on the table-type of ITP?

Minor things: English should be moderately edited.

Author Response

Point 1: The first general concern, is the fact that the blood is diluted to perform experiments, even when the same concentration of platelets is set as the dilution justification, the dilution factor is not comparable between patients, and thus, the final sample might be altered by that (and responses), considering a different dilution of RBCs and plasma in the mixture... Experiments should be done with whole blood, or reconstituted blood (to have the same amount of platelets per volume unit), or with washed platelets, with monitored plasma presence, etc. 

Response 1: General comment: We are grateful for the supportive comments and important suggestions regarding our manuscript. In the Revised version, we included GPVI-induced platelet activation into the model, and performed additional experiments to clarify the question of blood plasma concentration and platelet concentration in the tests. We were not able to perform some of the suggested experiments because, 1) for pediatric patients, the amount of blood available for analysis is very small (sometimes about one ml in total); 2) most of the analysed patients came for visits from other cities. Additionally, we should apply for a new Ethical committee permission to do such experiments as antibody determination.

Point 2: The first general concern, is the fact that the blood is diluted to perform experiments, even when the same concentration of platelets is set as the dilution justification, the dilution factor is not comparable between patients, and thus, the final sample might be altered by that (and responses), considering a different dilution of RBCs and plasma in the mixture... Experiments should be done with whole blood, or reconstituted blood (to have the same amount of platelets per volume unit), or with washed platelets, with monitored plasma presence, etc. 

Response 2: Thank you for raising this important point. To test this potential issue, we conducted preliminary experiments at the start of the study and demonstrated that the platelet responses are not significantly affected by the dilution. Illustration of this point is given in new Supplementary Figure S8. The dilution of 100 ul of leukocyte rich plasma per 1500 ul of Tyrode’s buffer was the lowest one utilised in the test (only for very few patients) and as could be seen from Fig. S8 it appears to activate platelets a little, but our preliminary experiments with inhibitors demonstrated that this activation originated from high platelet concentration leading to the secondary activation. This issue was demonstrated in our previous conference reports [42,71,72]. We have extended the description of the method to include this discussion.

Point 3: One of the main conclusions of this work, is that ITP platelets are preactivated, compared to healthy donor platelets. I have my doubts about this conclusion, since in Figure 1, it is shown that the SSC before stimulation with ADP (or agonist) is higher already in ITP platelets (wider range as well)... suggesting that actually, the integrity of ITP platelets is better preserved than those from healthy donors, which would also fit with a limited reaction capacity. The interpretation of basal Ca results are also a bit confusing, as graphs in Figure 1h do not entirely correspond with graphs in Figure 2d. Please discuss about this and reconsider the conclusion.

Response 3: We agree that SSC could be an indicator of granule status for objects of the same size. However, there is evidence that platelet size is increased in ITP [38], and we indeed observed that both FSC and SSC are increased and correlate with each other’s in our previous studies of platelets in ITP [9]. In the Revised Version, we clarify this interpretation in section 2.1 of the Results. In order to make results on basal calcium in Fig. 1h more clear and allow comparison with Fig. 2d, we added an inset showing the initial portion of the curve.

Point 4: Figure 2 shows the stratification of ITP platelets on two subgroups based on their ability to bind FBG, which coincides with a reduced change on SSC.

What is the basal SSC on the two subgroups of ITP?, is it already different as to explain a potential "pre-activation" and anergy in one of the subgroups? Or is it not altered? The second would also support the explanations given by the authors.

Figure 2c, the legend covers some of the points..

Response 4: Thank you! We have corrected the Figure 2 and added information on SSC values in the text (Results section 2.2, paragraph 2). Briefly, SSC-A values was around 8000 mfi for the ITP and around 7000 mfi for GT and Healthy controls. Therefore, the size of platelets is altered in most ITP patients in agreement with previous works cited above.

Point 5: The computational model goes beyond my evaluation capacity, but I would be cautious with some statements, such as "increased cytosolic calcium concentration in quiescent state of platelets, and its explanation - could be preactivated with antibodies, or permeability": it is not clear to me whether Ca2+ is actually increased, and the assumption of antibodies causing pre-activation is not proven at all, neither the permeability.

An experiment to try to prove this, would be to incubate healthy donor platelets with plasma/serum/Isolated Ig Fraction from the pediatric ITP patients, to see if that pre-activation could be induced.

Permeability could be assessed by permeable dies..

Response 5: Thank you for this suggestion. Although the measurement of calcium concentration in quiescent state of platelets proposed in this work has increased reliability because of direct comparison with EGTA-induced control values, we softened the statements in the manuscript to avoid ambiguity. By the term “permeability” we meant conductivity for calcium ions alone, and we corrected this issue throughout the text. Following your suggestion, we conducted preliminary experiments (n=3), where platelet free plasma from ITP patients was incubated with platelet rich plasma of healthy pediatric donors, and in one of three cases platelet pre-activation appeared (Fig. S6). However, further experiments should be conducted to investigate this point.

Point 6: While the stratification of ITP platelets, based on fibrinogen binding, was found (as said in the text) only significant with ADP, the ex vivo thrombus formation experiment is performed on flow chambers coated with collagen.

Still, in Supplementary, it looks as if stimulation with CRP does separate significantly those ITP subgroups based on FBG binding. Still, CRP seems to work through GPVI activation, and collagen might be recognized more efficiently by B1 integrin.

Please explain about this, and add Syk as another factor on the computational model.

Response 6: Thank you for this question. Although the fibrinogen binding level in response to CRP is more diverse in the ITP group than in the Healthy donors group, the cluster analysis did not find distinct clusters in response to CRP as stated in the Results section 2.1. The reason for this could originate from the variability of platelet responses to CRP [53] due to the highly variable number of GPVI receptors between donors [54], GPVI receptor polymorphysms [55,56] or other differences [53,57,58]. We included GPVI pathway into the computational model and demonstrated that the model HFB phenotype could become even more pronounced (Fig. S4b). We included discussion of this issue in the paragraph #3 of the Discussion.

Point 7: The flow assay could be done with fibrinogen coated chambers.

Another caveat of this assay is the fact that whole blood was assessed, with the difference on platelet counts. As mentioned before, it should be done with the same number of platelets per volume unit, which can be done by reconstituting blood (not diluting), or by performing the assay with washed platelets, with a specific plasma proportion.

Response 7: Thank you for this suggestion. We performed additional experiments with flow assay with blood with reduced platelet count to demonstrate its influence on thrombus growth. The methodology of the experiments is described in the Methods section. The flow assay with fibrinogen (when platelets do not form aggregates) is beyond our ability at this time, yet we mention its potential in the Discussion. We performed preliminary experiments with various proportions of plasma in the whole blood and determined that as low as 10% of plasma is enough for normal thrombus growth.

Point 8: The fact that we always have patient plasma on the samples that are tested, is a limitation on the interpretation of data (is something in the plasma interfering with results?, which results are platelet-dependent or endogenous to platelets?). This should be addressed by the authors in the disunion, and some example experiments should be performed to show the results obtained are platelet-dependent or not.

Response 8: Thank you for raising this question. We as well as others performed various checks on the influence of platelet concentration and plasma proteins on platelet reactivity [refs. 5,7,8,9,39,42,71,72] and still there is no conclusive data. We compared the results obtained from plasma dilutions used in this work (see Response 2) and did not find significant difference.

Point 9: I find absolutely necessary to provide data on autoantibodies in these patients, MAIPA testing or similar, to identify presence or not of B3 antibodies (vs antibodies against other glycoproteins). A correlation of results, with these valuable data, could provide with a better explanation of the findings.

Response 9: Thank you. We agree completely that these data would have greatly improved the study and we added this statement into the discussion. However, we did not expect this initially, and the Ethical Committee did not approve additional blood collection without a clear justification. Unfortunately, it was not possible for us to obtain these data now with another cohort of patients due to the national lockdown.

Point 10: Regarding the Supplementary data, the tables do not contain the platelet measuring unit. It would be great to have IPF on those samples, and the same table for the healthy donors. What is TPO on the table-type of ITP?

Response 10: Thank you for this suggestion. However, the data on patients provided by the clinical diagnostics lab do not include such information. The platelet count diagnostics was not performed for heathy donors and the amount of platelets was calculated in the blood samples by flow cytometry, which is not very accurate. Thus we do not provide this values as Supplementary Information. “TPO” means whether the patient received agonists of thrombopoietin receptor in the medical history, this information is now added as a footnote to Table S1.

Point 11: Minor things: English should be moderately edited..

Response 11: Thank you. We double-checked the English grammar in the Revised Version.

Reviewer 2 Report

The aim of the project was to analyze platelet integrin activation and calcium signaling in blood samples of pediatric patients with immune thrombocytopenia (ITP). The authors studied whole blood samples from pediatric IT patients, and healthy donor. They used continuous flow cytometry to analyze platelet responses and computational modeling. The data show that platelets from all ITP patients had significantly higher cytosolic calcium concentration in their resting state, as compared with healthy donors. Moreover, platelet shape change in response to ADP and fibrinogen varied significantly within ITP groups. A subgroup of ITP patients with low integrin activation altogether demonstrated phenotype similar to (Glandzmann thrombasthenia (GT) patients.  

Abstract: The novelty of the findings is not well unlined in the abstract. If I understand correctly, the technique used (continuous flow cytometry) is novel, also.

It is not clear why you use platelets from GT patients in the abstract. Please clarify.

The sample size of the patients seems quite small, in particular the healthy control. Were they sex matched? Did the authors analyze any correlation between sex and the low integrin activation group?  

I discourage the use of words like “slightly” or “somewhat higher” in the manuscript. I understand that non-statistically significant changes can still have biological relevance, however I would use a more appropriate language, such as: “increase although not statistically significant”.

Figure 3: a) this figure is truthful but very busy. I suggest simplifying it and including only necessary points. B) it is not clear what the “theory” is. Is that the control? To me it seems unnecessary. The “theory” can just be described in the text.

Figure 4: f) it is very confusing. I think it would be preferable to have a graph of means and S.E.M. and then submit the single values in a different file/format.   

Did the authors analyze any changes in signaling pathway(s)?

Table 1: I do not see any statistic.

Discussion: the discussion needs to be revised. The first paragraph is a repeat of the results. The second and third paragraphs can be combined. There is a lot of discussion about signaling but no data are provided to support it.  

Author Response

Point 1: The aim of the project was to analyze platelet integrin activation and calcium signaling in blood samples of pediatric patients with immune thrombocytopenia (ITP). The authors studied whole blood samples from pediatric IT patients, and healthy donor. They used continuous flow cytometry to analyze platelet responses and computational modeling. The data show that platelets from all ITP patients had significantly higher cytosolic calcium concentration in their resting state, as compared with healthy donors. Moreover, platelet shape change in response to ADP and fibrinogen varied significantly within ITP groups. A subgroup of ITP patients with low integrin activation altogether demonstrated phenotype similar to (Glandzmann thrombasthenia (GT) patients. 

Response 1: Thank you for your kind and encouraging evaluation of our manuscript. In the revision, we did our best to include the missing data, perform new calculations and modify the text as you have recommended.

Point 2: Abstract: The novelty of the findings is not well unlined in the abstract. If I understand correctly, the technique used (continuous flow cytometry) is novel, also.

Response 2: Thank you. We rephrased the abstract to underline the novelty of the method.

Point 3: It is not clear why you use platelets from GT patients in the abstract. Please clarify.

Response 3: Thank you for this question. We clarified in the abstract now that the GT group was supposed to be used as negative control for platelet integrin functioning.

Point 4: The sample size of the patients seems quite small, in particular the healthy control. Were they sex matched? Did the authors analyze any correlation between sex and the low integrin activation group? 

Response 4: Thank you for this suggestion. We have added three new healthy donors into the control group. Although the healthy donors were not sex-matched with the ITP patients, we had comparable number of males and females in both groups. This information is included in the Methods section now. The integrin binding does not appear to correlate with sex: we performed this analysis and added information into the Results section 2.2.

Point 5: I discourage the use of words like “slightly” or “somewhat higher” in the manuscript. I understand that non-statistically significant changes can still have biological relevance, however I would use a more appropriate language, such as: “increase although not statistically significant”.

Response 5: Thank you for advice! We softened the statements where possible.

Point 6: Figure 3: a) this figure is truthful but very busy. I suggest simplifying it and including only necessary points. B) it is not clear what the “theory” is. Is that the control? To me it seems unnecessary. The “theory” can just be described in the text.

Response 6: Thank you for raising this point. We have removed the model validation data into the SI now with information on it in the Methods section of the Revised version.

Point 7: Figure 4: f) it is very confusing. I think it would be preferable to have a graph of means and S.E.M. and then submit the single values in a different file/format.  

Response 7: Thank you. We removed the panel 4f to the SI (new panel S5g) in preference of the new data. The values on neutrophil velocity are included in the main text now.

Point 8: Did the authors analyze any changes in signaling pathway(s)?

Response 8: Thank you for this suggestion. We had analysed mostly changes in calcium-related enzymes. In the new version, we demonstrate that the reduced amount of Rap1 (or other signaling proteins between calcium and integrins) will lead to the reduced amount of active integrins (Fig. 3b). We included this information in Discussion as well. 

Point 9: Table 1: I do not see any statistic.

Response 9: Thank you for this suggestion. We have included the statistical significance data in the Table 1.

Point 10: Discussion: the discussion needs to be revised. The first paragraph is a repeat of the results. The second and third paragraphs can be combined. There is a lot of discussion about signaling but no data are provided to support it.. 

Response 10: Thank you for your advice. We removed the majority of the first paragraph, and shortened and combined the second one with the third one. In the remaining part, we shortened the sections not sufficiently supported by data

Round 2

Reviewer 1 Report

I find the reviewed version of the manuscript has improved considerably and the authors have addressed all the points raised, and acknowledge the limitations on its present state. The reviewer is confident the authors will address the remaining questions in future studies.

Reviewer 2 Report

The authors properly addressed all the reviewer's points. Good luck with your future work!